# KLF5 Is Crucial for Androgen-AR Signaling to Transactivate Genes and Promote Cell Proliferation in Prostate Cancer Cells

**DOI:** 10.3390/cancers12030748

**Published:** 2020-03-21

**Authors:** Juan Li, Baotong Zhang, Mingcheng Liu, Xing Fu, Xinpei Ci, Jun A, Changying Fu, Ge Dong, Rui Wu, Zhiqian Zhang, Liya Fu, Jin-Tang Dong

**Affiliations:** 1Department of Genetics and Cell Biology, College of Life Sciences, Nankai University, 94 Weijin Road, Tianjin 300071, Chinafuchu12@nankai.edu.cn (L.F.); 2School of Medicine, Southern University of Science and Technology, 1088 Xueyuan Road, Shenzhen, Guangdong 518055, China; zhangzq@sustc.edu.cn; 3Emory Winship Cancer Institute, Department of Hematology and Medical Oncology, Emory University School of Medicine, 1365-C Clifton Road, Atlanta, GA 30322, USA; 4Vancouver Prostate Centre, Department of Urologic Sciences, University of British Columbia, Vancouver, BC V6H 3Z6, Canada

**Keywords:** KLF5, androgen receptor, cell proliferation, tumorigenesis, prostate cancer

## Abstract

Androgen/androgen receptor (AR) signaling drives both the normal prostate development and prostatic carcinogenesis, and patients with advanced prostate cancer often develop resistance to androgen deprivation therapy. The transcription factor Krüppel-like factor 5 (KLF5) also regulates both normal and cancerous development of the prostate. In this study, we tested whether and how KLF5 plays a role in the function of AR signaling in prostate cancer cells. We found that KLF5 is upregulated by androgen depending on AR in LNCaP and C4-2B cells. Silencing *KLF5*, in turn, reduced AR transcriptional activity and inhibited androgen-induced cell proliferation and tumor growth in vitro and in vivo. Mechanistically, KLF5 occupied the promoter of *AR*, and silencing *KLF5* repressed *AR* transcription. In addition, KLF5 and AR physically interacted with each other to regulate the expression of multiple genes (e.g., *MYC*, *CCND1* and *PSA*) to promote cell proliferation. These findings indicate that, while transcriptionally upregulated by AR signaling, KLF5 also regulates the expression and transcriptional activity of AR in androgen-sensitive prostate cancer cells. The KLF5-AR interaction could provide a therapeutic opportunity for the treatment of prostate cancer.

## 1. Introduction

Prostate cancer (PCa) is prevalent among older men; and is one of the common causes of cancer-related death in men. While genetic and epigenetic alterations of multiple genes, including loss of *PTEN* [1,2,3], fusion between *TMPRSS2* and *ERG* [4,5], amplification and over-expression of *MYC*, and inactivation of *P53* and *RB* [3], initiate and promote prostatic carcinogenesis [6,7,8], androgen/androgen receptor (AR) signaling is the driving force in the process [9,10]. AR is thus a major therapeutic target, and androgen deprivation therapy (ADT) via surgical or chemical castration, including abiraterone and enzalutamide treatment, is thus the most commonly used effective therapy for patients with PCa. Unfortunately, PCa often develop resistance to ADT and become castration-resistant prostate cancers (CRPCs), which usually maintain AR activity by different mechanisms, such as generating AR splice variants, gain-of-function mutations in *AR*, and functional alterations leading to androgen independence [11,12,13].

AR is a member of the nuclear steroid receptor superfamily that is predominantly activated by testosterone and di-hydrotestosterone [14,15]. AR signaling is essential not only for postnatal development and maintenance of normal prostates but also for the regeneration of prostates after androgen deprivation. AR signaling also promotes the development and progression of PCa via enhanced cell proliferation and survival [16]. Many PCa driver genes alter the activity or structure of AR or are regulated by AR signaling during prostatic carcinogenesis.

KLF5 is a basic transcription factor that belongs to the Krüppel-like factor (KLF) family. It regulates a variety of biological processes including cell proliferation, apoptosis, angiogenesis, stemness and the epithelial-mesenchymal transition (EMT) [17,18]. KLF5 also functions in multiple pro- and anti-proliferative signaling pathways, including the RAS/ERK and PI3K/AKT proliferative pathways and the TGF-β anti-proliferative signaling to regulate different cancer cell behaviors [19,20,21]. As a transcription factor, KLF5 interacts with other transcription factors such as c-Jun [22], p53 [23], and ERα [24] to regulate the transcription of many genes involved in cell proliferation and tumorigenesis [25], including *CCND1* and *MYC* [26,27,28]. In the prostate, KLF5 also plays crucial roles in postnatal development, regeneration after castration, and PCa. In both human and mouse prostates, Klf5 is expressed in both basal and luminal cells, and basal cells preferentially express acetylated Klf5 [29,30]. Androgen ablation by castration in mice increases both Klf5 expression level and the number of KLF5-expressing cells [29], and both Klf5 and acetylated Klf5 are indispensable for the maintenance of basal progenitors and their luminal differentiation [30]. Klf5 and its acetylation are also necessary for the survival and regeneration of basal progenitor-derived luminal cells following castration and subsequent androgen restoration [30]. During tumorigenesis, the deletion of *Klf5* promotes *Pten* loss-induced prostate tumors, and the *Klf5*^-/-^/*Pten*^-/-^ tumors also have increased basal to luminal differentiation [31]. 

Taken together with the facts that androgen/AR signaling is the driving force in both normal prostate development and regeneration and PCa development, both KLF5 and AR are transcription factors, and androgen appears to induce the expression of *KLF5* in PCa cells [32,33], we propose that KLF5 and AR could be functionally associated with each other in prostatic carcinogenesis. We tested this hypothesis in this study. We demonstrated that silencing *KLF5* inhibited cell proliferation and tumor growth of PCa cells. In addition, as a transcription factor, KLF5 occupied the promoter of *AR* to promote its transcription; and KLF5 was also required for AR’s transcriptional activity. Furthermore, KLF5 and AR interacted with each other to regulate transcription of AR target genes (e.g., *MYC*, *CCND1,* and *PSA*) to promote cell proliferation and tumor growth. These findings suggest that specific targeting of the AR-KLF5 interaction could be a potential therapeutic strategy for disrupting androgen signaling in PCa treatment.

## 2. Results

### 2.1. Androgen/AR Signaling Upregulates KLF5 Transcription in PCa cells

To test the role of androgen/AR signaling in KLF5 transcription, we measured KLF5 expression in two androgen-responsive PCa cell lines, LNCaP and C4-2B, in hormone-free medium (RIPA1640 medium supplemented with charcoal-stripped bovine fetal serum) treated with varying concentrations of R1881, a synthetic androgen, specifically binds to AR with higher affinity than dihydrotesterone (DHT), and R1881-bound AR dimerizes and translocates to the nucleus to interact with coregulators to regulate gene transcription [34,35]. Androgen treatment caused a dose-dependent increase in KLF5 expression at both protein and mRNA levels (Figure 1a–d). As expected, R1881 treatment also increased the expression of known AR targets *PSA*, *TMPRSS2*, and *FKBP5* at the mRNA level (Figure 1b). Treatment of C4-2B cells with R1881 at 10 nM for different times increased KLF5 expression in a time-dependent manner (Figure 1e,f). 

The same two cell lines grown in normal medium, which contains hormones to activate AR signaling, were treated with enzalutamide to block androgen/AR signaling. Enzalutamide is an AR antagonist that binds with AR to block its nuclear translocation and subsequent interactions with its coactivators in regulation target gene transcription [36,37]. Enzalutamide is widely used in the treatment of PCa [38,39]. Enzalutamide treatment at varying concentrations caused a dose-dependent decrease in KLF5 expression at both protein and mRNA levels in both cell lines (Figure 1g–j), while decreasing the expression of AR and its target genes (*PSA*, *TMPRSS2*, and *FKBP5*) as expected (Figure 1h). Consistently, treatment of C4-2B cells with 10 µM enzalutamide for different times decreased the expression of both KLF5 and AR in a time-dependent manner (Figure 1k,l).

To test whether R1881-induced KLF5 expression depends on AR, we knocked down AR by siRNA in C4-2B cells cultured in hormone-free medium in the presence of R1881 (10 nM) or enzalutamide (10 µM) and analyzed KLF5 expression. AR silencing by siRNA, which was confirmed by western blotting (Figure 1m), eliminated the induction of KLF5 by R1881 at both protein and mRNA levels (Figure 1m,n). Although cells were cultured in hormone-free medium, enzalutamide treatment still reduced AR protein (Figure 1m) and KLF5 mRNA levels (Figure 1n). Further supporting a role of AR in R1881-induced KLF5 expression, the promoter-luciferase reporter assay demonstrated that, in C4-2B cells cultured in hormone-free media, R1881 induced a significant KLF5 promoter activity while enzalutamide decreased the activity, and AR silencing eliminated these effects (Figure 1o). These findings indicate that androgen/AR signaling upregulates KLF5 transcription.

### 2.2. KLF5 is Crucial for Maintaining the Transcriptional Activity of AR in PCa Cells

To determine whether androgen-upregulated KLF5 has a functional role in androgen/AR signaling, we evaluated whether knockdown of KLF5 affects AR’s transcriptional activity in LNCaP cells with KLF5 silencing by siRNA and C4-2B cells with KLF5 silencing by shRNA. Cells were cultured in regular media, which contains hormones as regular FBS was used. Enzalutamide treatment was applied to inhibit AR signaling activity. KLF5 silencing clearly decreased the expression of PSA, a classic transcriptional target of AR [40], at both protein and mRNA levels in both LNCaP and C4-2B cell lines (Figure 2a,b). With enzalutamide treatment, expression of both PSA and AR was dramatically reduced, and the effect of KLF5 knockdown on PSA expression was weakened at the protein and mRNA level (Figure 2a–d). The mRNA expression of two additional AR target genes, *TMPRSS2* and *FKBP5* [41,42], was also decreased by KLF5 silencing, as detected by real-time qPCR (Figure 2c,d). We noticed that KLF5 silencing also decreased AR expression in both cell lines (Figure 2a,b), which is further addressed in Figure 3.

We also analyzed the activities of two androgen-responsive promoters, *PSA* and *MMTV* [43], in the same cells with the same treatments. The activities of both *PSA* and *MMTV* promoters were significantly decreased by KLF5 knockdown, and enzalutamide treatment eliminated both the promoter activities and the effect of KLF5 silencing on promoter activities (Figure 2e,f). 

To test whether KLF5 directly binds to the promoters and enhancer of AR target genes, we performed ChIP assay using both AR and KLF5 antibodies. In AR-precipitated DNA, the *PSA* and *FKBP5* promoters and the *TMPRSS2* enhancer were detected by PCR in C4-2B cells, as expected, and KLF5 silencing reduced the promoter and enhancer DNA (Figure 2g). In KLF5-precipitated DNA, while the promoter DNA of *PSA* was detected, neither the *FKBP5* promoter nor the *TMPRSS2* enhancer was detected (Figure 2g), and the *PSA* promoter was eliminated by the inhibition of AR signaling by enzalutamide (Figure 2h). These results suggest that KLF5 is crucial for the transcriptional activity of AR.

### 2.3. KLF5 also Promotes Transcription of the AR Gene in PCa Cells

Analyzing the effect of KLF5 on AR’s gene transactivating function, we noticed that knockdown of KLF5 reduced AR expression in both LNCaP and C4-2B cells (Figure 2a,b). We thus tested whether KLF5 modulates the expression of AR. Knockdown of KLF5 by siRNA in LNCaP cells or by shRNA in C4-2B cells clearly reduced AR expression at both protein and mRNA levels (Figure 3a–d).

To further test whether KLF5 directly promotes AR transcription, we analyzed the 2-Kb immediate AR promoter sequence for potential KLF5 binding sites using the JASPAR database, in which the consensus KLF5 binding sequences were defined by ChIP-Seq study [40]. Multiple such sites were predicted, and the 5 with the highest binding scores were all located within the immediate 350-bp AR promoter (Figure 3e). 

We then constructed a promoter-luciferase reporter plasmid with the entire 2-Kb immediate AR promoter sequence in the pGL3 vector (pGL3-AR, Figure 3f). Two additional AR promoter luciferase reporter plasmids were also constructed, one with the upper 1.5-Kb and the other with the lower 0.5-Kb of the full 2-kb promoter sequence (pGL3-AR1 and pGL3-AR2, Figure 3f). Both pGL3-AR and pGL3-AR2 also contained 0.2-Kb sequence of exon 1 (Figure 3f). Interestingly, knockdown of KLF5 significantly decreased the activities of pGL3-AR and pGL3-AR2, both of which contained potential KLF5 binding sites, but not that of pGL3-AR1, which did not contain a KLF5 binding site (Figure 3g). Therefore, it is likely that KLF5 directly promotes AR transcription via promoter binding.

To test whether KLF5 directly binds to AR promoter, ChIP was performed with KLF5 antibody in C4-2B cells and PCR performed with primers to amplify the AR promoter in pGL3-AR2 in three fragments, A, B, and C (Figure 3f). Fragment B, which contained three potential KLF5 binding sites (Figure 3f), was detected in the DNA pulled down by KLF5 antibody, but fragments A and C were not (Figure 3h). Further analysis showed that fragment B2, containing the sites from −107 to −93, was detected in the DNA pulled down by KLF5 antibody, but fragment B1 was not. Therefore, KLF5 can directly bind the AR promoter via the sites from −107 to −93.

### 2.4. KLF5 Physically Associates with AR in Prostate Epithelial Cells

Considering that KLF5 is crucial for AR function and that both KLF5 and AR are transcription factors, it is likely that KLF5 and AR could physically associate with each other to regulate gene transcription. We transfected FLAG-tagged KLF5 (Flag-KLF5) with pSG5-AR or FLAG-tagged AR (Flag-AR) with HA-tagged KLF5 (HA-KLF5) into HEK293T cells, and performed IP with anti-Flag antibody. Western blotting detected AR in the KLF5 precipitate (Figure 4a) and KLF5 in the AR precipitate (Figure 4b), supporting a physical association between KLF5 and AR. We noticed that enzalutamide treatment reduced AR in the KLF5 precipitate (Figure 4c), which could suggest a role of AR’s ligand binding domain in the AR-KLF5 interaction.

We also divided KLF5 into two fragments, one with residues 1–200 and the other with 201–453, and tested which fragment mediates KLF5′s association with AR using the same approaches as in Figure 4a,b. The domain of KLF5 mediating the KLF5-AR interaction was restricted residues 1 to 200 (Figure 4d). 

We also performed IP with anti-KLF5 or anti-AR antibody to pull down their respective protein complexes in C4-2B cells. Western blotting detected AR in the KLF5 complex and KLF5 in the AR complex (Figure 4e,f), indicating a physical association between the endogenous KLF5 and AR. This set of experiments was repeated with mouse prostate lysates, and the endogenous Ar-Klf5 association was again detected (Figure 4g,h). Therefore, KLF5 physically associates with AR in prostate cells.

### 2.5. KLF5 Is also Crucial for AR-Mediated MYC and Cyclin D1 Expression in PCa Cells

In PCa cells, AR promotes cell proliferation via the upregulation of a subset of genes such as *MYC* and *CCND1*, and KLF5 has also been shown to upregulate the same two genes in epithelial cells [26,27,28,44,45,46,47]. We thus tested whether KLF5 is also required for AR to upregulate *MYC* and *CCND1* in androgen-responsive PCa cells. In LNCaP and C4-2B cells cultured in normal medium, knockdown of KLF5 decreased the expression MYC and cyclin D1 at both protein (Figure 5a,b) and mRNA levels (Figure 5c,d). When AR activity was inhibited by enzalutamide at 10 µM for 24 h, both MYC and cyclin D1 were significantly downregulated, and silencing KLF5 had little or no effect on the expression of MYC and cyclin D1 (Figure 5a–d). ChIP-PCR demonstrated that the amount of AR bound to the promoters of *CCND1* and *MYC* was apparently reduced by the knockdown of KLF5 (Figure 5e), and similarly, the amount of KLF5 bound to the same two promoters was also reduced by inhibiting AR signaling with enzalutamide treatment in C4-2B cells (Figure 5f).

### 2.6. KLF5 Is Crucial for Androgen/AR to Promote Cell Proliferation and Tumor Growth in PCa Cells

Based on the necessity of KLF5 for AR to regulate genes including *MYC* and *CCND1*, we tested whether KLF5 is indeed involved in the pro-proliferative function of AR in PCa cells. In LNCaP or C4-2B cells, colony and sphere formation assays demonstrated that silencing KLF5 by RNAi in normal medium significantly reduced colony-forming efficiency in 2-D culture (Figure 6a,b) and sphere formation in Matrigel (Figure 6c,d). Inhibition of AR signaling by enzalutamide treatment strongly suppressed both colony and sphere formation (Figure 6a–d). Further, under the condition of AR inhibition, KLF5 silencing had a weaker yet detectable effect (Figure 6a–d).

We also tested whether KLF5 is necessary for AR to promote xenograft tumorigenesis. C4-2B cells with stable knockdown of KLF5 were inoculated into immunosuppressed female BABL/c nude mice, with or without enzalutamide treatment (10 mg/kg, administered via oral gavage once a day for up to 21 days, for the tumorigenesis assays). Consistent with colony and sphere formation results, KLF5 silencing reduced tumor growth, as indicated by tumor images and tumor weights at excision (Figure 6e,f). Enzalutamide treatment also reduced tumor growth, and KLF5 silencing and enzalutamide had an additive effect on tumor growth (Figure 6e,f). 

In the tumor xenografts, IHC staining demonstrated enzalutamide treatment reduced the expression of both KLF5 and AR, which is consistent with in vitro findings (Figure 1g–l), and KLF5 silencing also downregulated AR expression (Figure 6g,h). Similarly, IHC staining demonstrated that both KLF5 silencing and enzalutamide treatment reduced the expression of Ki67, a cell proliferation marker; the number of Ki67-positive cells; and the expression of cyclin D1 and MYC (Figure 6i,j). These results indicate that KLF5 is crucial for AR to function in PCa cells.

## 3. Discussion

Our findings in this study indicate that KLF5 is crucial for androgen/AR signaling to function in PCa cells. The first line of evidence is that the transcriptional activity of AR depended on the expression of KLF5. For example, *KLF5* silencing decreased the expression of *PSA*, a classic transcriptional target gene of AR in the prostate [41], and *TMPRSS2* and *FKBP5*, two other AR target genes [42,43], in LNCaP and C4-2B cells (Figure 2). The necessity of KLF5 for AR’s transcriptional activity was also demonstrated by promoter luciferase reporter assays using two androgen responsive promoters (i.e., *PSA* and *MMTV* [48]) (Figure 2) and by the expression of two genes that mediate AR’s pro-proliferative function, i.e., *MYC* and *CCND1* [26,27,28,44,45,46,47], in the same cells with *KLF5* silencing (Figure 5). We have also presented evidence from cellular analyses, in which androgen/AR signaling also required KLF5 to maintain a steady proliferation of PCa cells. Specifically, *KLF5* silencing significantly attenuated the functions of AR in the maintenance of colony and sphere formation in vitro and xenograft tumor growth in nude mice (Figure 6a). Consistently, *KLF5* silencing also reduced the number of Ki67-positive cells and the expression of cyclin D1 and MYC (Figure 6). We noticed that, after blocking AR activity with enzalutamide, which had a profound effect (Figure 5 and Figure 6), *KLF5* silencing still had a detectable effect in both the expression of MYC and cyclin D1 and cell proliferation (Figure 5 and Figure 6), which suggests that, while crucial for AR to function, KLF5 can still function when AR is inhibited. Indeed, in androgen-independent PCa cell lines including PC-3 and DU 145, KLF5 is clearly pro-proliferative, even though when TGF-β is activated, TGF-β and KLF5 slow but do not stop cell proliferation [49,50].

Molecularly, the enhancing effect of KLF5 on AR function in cell proliferation is mediated by at least three distinct mechanisms. For example, via direct promoter binding, AR promotes the transcription of *KLF5* to increase its expression [32]. As expected, the upregulation of *KLF5* by androgen was mediated by AR (Figure 1), since inhibition of AR by RNAi-mediated *AR* silencing or enzalutamide treatment eliminated the induction of *KLF5* transcription (Figure 1).

The second molecular mechanism by which KLF5 facilitates AR function is that KLF5 also activates *AR* transcription in PCa cells. For example, *KLF5* silencing by RNAi reduced AR expression in both LNCaP and C4-2B cells (Figure 3). In addition, the *AR* promoter indeed contained multiple consensus KLF5 binding elements that were necessary not only for the *AR* promoter’s activities in the promoter-reporter assay and but also for the binding of KLF5 to the *AR* promoter in the ChIP-PCR analysis (Figure 3), and two adjacent KLF5 binding elements in the *AR* promoter have been confirmed to be essential for KLF5 binding and promoter activity (Figure 3).

The third mechanism is that KLF5 and AR coordinate to regulate gene transcription, which is supported by multiple lines of evidence. Firstly, KLF5 and AR depend on each other in their binding to the promoters of *PSA*, *MYC* and *CCND1*, as *KLF5* silencing reduced the amount of promoter/enhancer DNA of these genes in AR-precipitated DNA (Figure 2) while inhibition of AR reduced this DNA in KLF5-precipitated DNA (Figure 5). Nevertheless, the details of KLF5 and AR binding to gene promoters are unclear (e.g., the chromatin landscape for the binding). Secondly, KLF5 and AR, both of which are transcription factors, physically associate with each other to regulate gene transcription. AR was detected in the KLF5 protein complex and KLF5 in the AR complex (Figure 4). In addition, the KLF5-AR interaction occurred not only for ectopically expressed KLF5 and AR in HEK293T cells but also for endogenous KLF5 and AR in both human cells and mouse prostates (Figure 4). Furthermore, the KLF5-AR interaction was mediated by a sequence within residues 1-200 of KLF5 (Figure 4) and was attenuated by enzalutamide treatment (Figure 4). 

Therefore, androgen/AR signaling activates *KLF5* expression via the binding of AR to *KLF5* promoter, KLF5 in turn enhances the transcription of *AR* by promoter binding, and AR and KLF5 then coordinate to transactivate a subset of genes to promote the proliferation of PCa cells.

We noticed that, for AR target genes *PSA*, *TMPRSS2,* and *FKBP5*, while *KLF5* silencing reduced their induction by AR (Figure 2a–d), which supports the necessity of KLF5 for AR function in their transcription, AR-bound promoter DNA was detected in KLF5-precipitated promoter DNA only for *PSA* but not for *TMPRSS2* and *FKBP5* (Figure 2). The reason for this discrepancy is unknown. Neither is it known whether the KLF5-AR association depends on promoter DNA or cofactors of AR.

In mouse prostates, castration-mediated androgen depletion increased Klf5-positive cells [29], which is seemingly inconsistent with the induction of KLF5 by androgen in PCa cells (Figure 1). Compared to luminal cells in the prostate, basal cells preferentially express Klf5, particularly acetylated Klf5 [30], castration causes massive death in luminal cells but much less so in basal cells, and basal cells express much less AR and are androgen insensitive. LNCaP and C4-2B cells are AR-positive and androgen-dependent/sensitive, and thus have a different lineage from basal cells.

The role of KLF5 in androgen-induced cell proliferation and tumor growth could also involve KLF5′s function in tumor microenvironment (TME) and immune responses. For example, pro-inflammatory TNFα and lipopolysaccharide (LPS) induce KLF5 expression, and TNFα depends on KLF5 to induce MCP-1 [51]. In addition, KLF5 directly interacts with NF-κB [52], a potent inflammatory factor, and interruption of this interaction inhibits LPS-induced macrophage proliferation [53]. Knockdown of KLF5 also reduces the expression of p50 and p65 subunits of NF-κB and its downstream target genes TNFα and IL-6 in response to LPS [54]. This and other potential mechanisms for KLF5 function remain to be examined.

During late stages of tumor progression, AR becomes activated even when androgen levels are low, causing CRPC [9,10]. Further studying how KLF5 and AR coordinate to regulate the expression of genes, particularly those mediating cell proliferation/survival and thus likely affecting PCa progression, will facilitate our understanding of AR activation in CRPC and the development of therapeutic approaches for the treatment of CRPCs. For example, in advanced PCa, the *KLF5* locus often undergoes hemizygous deletion [21,55], which downregulates *KLF5* expression because the gene is haploinsufficient [31]. As KLF5 is necessary for AR function in PCa cells, as discussed above, downregulation of KLF5 could generate a feedback signal that leads to the upregulation of AR or functional compensation of AR activity. This hypothesis remains to be tested.

## 4. Materials and Methods 

### 4.1. Cell Lines, Cell Culture, and RNA Interference

Human PCa cell line LNCaP was purchased from American Type Cell Culture (Manassas, VA) and cultured in RPMI-1640 medium supplemented with 10% fetal bovine serum (FBS, Gibco, Waltham, MA). The C4-2B cell line, originally derived from a bone metastasis of a LNCaP clone in mice [56], was kindly provided by Dr. Leland W. K. Chung of Cedars-Sinai Medical Center and cultured in the same medium as LNCaP. Cells were maintained at 37 °C with 5% CO_2_. During experiments, cells recovered from a liquid nitrogen freezer were used within two months (<20 passages) with no noticeable morphological changes. All cell lines were authenticated by STR profiling before experiments were started. For all experiments involving R1881 (Melonepharma, Dalian, China, catalog number: MB5484) treatments, the medium was replaced with phenol red-free medium containing 10% charcoal-stripped FBS. 

For RNA interference (RNAi) with shRNA, C4-2B cells were infected with lentiviruses expressing an shRNA specifically targeting human *KLF5*, which was developed and validated in a previous study with various PCa cell lines [57], to establish the cell population in which *KLF5* is stably knocked down. For RNAi with siRNAs, siRNA (si*AR*: 5′-CAAGGGAGGUUACACCAAA-3′; si*KLF5*: 5′-AAGCUCACCUGAGGACUCA-3′) oligos against human *KLF5* and *AR* were synthesized by Sagon Biotech (Guangzhou, China) and transfected into cells using the Lipofectamine RNAiMAX reagent (Invitrogen, Carlsbad, CA) according to the manufacturer’s protocol.

### 4.2. Western Blot Analysis

Cultured cells were lysed in lysis buffer (50 mM Tris pH 7.4, 150 mM NaCl, 2 mM EDTA, 0.8% NP-40, 0.2% Triton X-100, 3% glycerol). Cell lysates were subjected to polyacrylamide gel electrophoresis (PAGE), and proteins were transferred onto a polyvinylidene fluoride (PVDF) membrane. Membranes were soaked in 5% nonfat milk or 5% BSA solution for one hour to block nonspecific binding of proteins, and then incubated with primary antibodies overnight at 4 °C. On the following day, membranes were incubated with secondary antibodies for 2 h at room temperature, and WesternBright ECL (Advansta, Menlo Park, CA) was used with the luminescent image analyzer (Jun Yi Dong Fang, Beijing, China) to capture images. Uncropped scans can be found in Appendix A.

Antibodies used in western blotting were: KLF5 (1:1000, 21017-1-AP, Proteintech), AR (1:1000, 5153S, Cell Signaling), PSA (1:3000, 10679-1-AP, Proteintech), MYC (1:1000, 9402, Cell Signaling), cyclin D1 (1:10000, ab134175, Abcam), FLAG (1:3000, SAB4200071, Sigma), HA (1:3000, 3724S, Cell Signaling), GAPDH (1:3000, 60004-1-Ig, Proteintech).

### 4.3. RNA Extraction and Real-time qPCR

Total RNA was extracted from cells using the Trizol reagent (Invitrogen) and 2 μg total RNA reverse-transcribed using the PrimeScript™ RT reagent Kit with gDNA Eraser (TaKaRa, Tokyo, Japan). Real-time qPCR was performed with the SYBR Green MasterMix reagent (Takara) using the Mastercycler Realplex real time PCR system (Eppendorf, Hamburg, Germany). Human *GAPDH* gene served as an internal control. The comparative 2^-^^△△Ct^ method was used to calculate gene expression levels. Each sample was analyzed in triplicate. Primer sequences for real-time qPCR are listed in Appendix A.

### 4.4. Construction of Expression and Luciferase Report Plasmids

Mammalian expression plasmids for Flag-KLF5, HA-KLF5, pSG5-AR, Flag-AR, and luciferase reporter plasmids for pGL3-KLF5, pGL3-PSA, and pGL3-MMTV were generated in our laboratory. For promoter-luciferase plasmids, pGL3-AR, pGL3-AR1, and pGL3-AR2, primers were respectively designed with the entire 2-Kb immediate AR promoter sequence (AR), the upper 1.5-Kb (AR1) and the lower 0.5-Kb (AR2). PCR and cloning of PCR products were used to generate luciferase reporter plasmids for AR, AR1, and AR2 in pGL3-Basic vector following standard procedures. Primer sequences for gene cloning are listed in Appendix A.

### 4.5. Luciferase Reporter Assay

LNCaP and C4-2B cells were transiently transfected with pGL3-Basic, pGL3-KLF5, or pGL3-PSA, or pGL3-MMTV and pRL-TK (Renilla luciferase, Promega, Madison, WI) as an internal control. After 48 h of transfection and enzalutamide (Beyotime Biotechnology, Shanghai, China, catalog number: SC0074) treatment, cells were lysed in 5 × lysis buffer (Promega) for 30 min and luciferase activity was measured using a luminometer (Tristar LB941, Berthold Technologies, BadWild, Germany). Firefly luciferase activity was normalized to Renilla luciferase activities in each reaction. Experiments were performed in triplicate.

### 4.6. Colony Formation Assay

One thousand C4-2B cells/well or 2000 LNCaP cells/well were seeded in 6-well plates and cultured in the presence of RPMI-1640 medium containing 10% FBS with DMSO or 10 μM enzalutamide (Beyotime Biotechnology,). The plates were incubated for 2 (C4-2B) or 3 (LNCaP) weeks, after which cultures were fixed with 4% paraformaldehyde for 30 min and stained by 0.05% crystal violet (BBI life sciences, Shanghai, China) for 1 h at room temperature and then photographed. In a single experiment, assays were conducted in triplicate and then as three independent experiments.

### 4.7. 3D Matrigel Assay

In 8-well chamber slides (BD Bioscience, Shanghai, China, catalog number: 354108), 30 µL growth factor reduced BD Matrigel (BD Biosciences, catalog number: 354230) was added per well. Slides were placed in a cell culture incubator for at least 15 min to solidify the Matrigel. Next, 2000 C4-2B cells/well or 4000 LNCaP cells/well were seeded in RPMI-1640 medium containing 10% FBS with DMSO or 10 μM enzalutamide (Beyotime Biotechnology) and 2% Matrigel. Media were replenished every 3 days. Chamber slides were incubated for 2 (C4-2B) or 3 (LNCaP) weeks, and then photographed. Image J program (NIH, USA) was used to measure the diameter of each sphere. Spheres with a diameter larger than 80 μm were counted.

### 4.8. Immunoprecipitation

For exogenous immunoprecipitation, HEK293T cells were cotransfected with pcDNA3.0-Flag or Flag-KLF5 in the same vector with pSG5-AR plasmid using the Lipofectamine 2000 Transfection Reagent (Invitrogen) according to the manufacturer’s protocol. After 48 h, the cells were collected and resuspended in cell lysis buffer containing 50 mM Tris (pH 7.4), 150 mM NaCl, 2 mM EDTA, 0.8% NP-40, 0.2% Triton X-100, and 3% glycerol. EDTA-free Protease Inhibitor Cocktail (Roche, Indianapolis, IN) and PMSF (Sangon Biotech) were added to the cell lysis buffer. Cell lysates were incubated with anti-FLAG-agarose beads at 4 °C for 2 h (Sigma). Beads were washed and eluted and supernatants analyzed by western blotting. For endogenous immunoprecipitation, C4-2B cells or mouse prostate were collected and resuspended in lysis buffer and then rabbit or mouse normal IgG, KLF5 antibody (self-made) or AR antibody (06-680-AF488, Millipore) added to the lysate, followed by overnight incubation at 4 °C. Lysates coupled with antibody were incubated with magnetic Dynabeads (Invitrogen) for 2 h. Beads were washed extensively and eluted and supernatants analyzed by western blotting as above.

### 4.9. Mouse Xenograft Studies

C4-2B (2×10^7^) cells in 100 µL PBS-Matrigel (1:1) (BD Biosciences, catalog number: 354248) were implanted via subcutaneous injection into the flanks of mice. Cells were left to grow for one week. Mice were randomly divided into control or enzalutamide (10 mg/kg) (MedChemExpress, New Jersey, USA, catalog number: HY-70002) [58] groups, which were given once a day via oral gavage for up to 21 days (n=6/group). Mice were euthanized and tumors were surgically dissected, immediately weighed and fixed in 10% formalin for standard histopathological evaluation.

### 4.10. Immunohistochemistry (IHC)

IHC staining was performed to detect protein expression of KLF5 (1:400, Proteintech), AR (1:800, 06-680-AF488, Millipore), Ki67 (1:2000, ab15580, Abcam), cyclin D1 (1:2000, ab134175, Abcam) and MYC (1:1000, 9106, Abcam) in tumor xenografts. Formalin-fixed paraffin embedded tissues were sectioned at 4 μm, deparaffinized in xylene, rehydrated in graded ethanol (100–75%) and repaired antigen by boiling the slides in a citrate buffer (10 mM trisodium citrate, pH 6.0) for 3 min using a pressure cooker. After treatment with 3% H_2_O_2_ for 10 min, tissue sections were blocked with 10% normal goat serum, incubated with primary antibodies at 4 °C overnight, and then with EnVision PolymerHRP secondary antibodies (MXB Biotechnologies, Fuzhou, China) at room temperature for 30 min. After chromogenic reaction using DAB (MXB Biotechnologies), nuclei were stained with hematoxylin (MXB Biotechnologies). Finally, tissue sections were dehydrated and mounted.

### 4.11. Chromatin Immunoprecipitation (ChIP)

C4-2B cells were grown for 3 days in RPMI-1640 medium supplemented with 10% FBS. ChIP was performed using the Simple ChIP Enzymatic Chromatin IP Kit (Cell Signaling Technology, Danvers, MA, catalog number: #9003) according to the manufacturer’s instructions. Firstly, cells were cross-linked with 1% formaldehyde for 10 min and quenched with glycine at room temperature. Samples were collected and digested using micrococcal nuclease for 20 min at 37 °C; reactions were stopped by the addition of 0.5 M EDTA and incubated in ChIP buffer with protease inhibitors on ice for 10 min. After sonication, chromatin extracts were immunoprecipitated using anti-AR (06-680-AF488, Millipore), anti-KLF5 (AF3758, R&D Systems) or anti-IgG antibody. ChIP products were detected by regular PCR (with input loading quantity of one fourth of IgG or AR) or real-time qPCR (each gene in triplicate). Sequences of PCR primers are described in Appendix A.

### 4.12. Statistical Analysis

All in vitro experiments were repeated at least three times. All numerical results are expressed as mean ± SD. Two group comparisons were compared using Student’s t test by SPSS 21 (IBM corporation, Armonk, NY, USA). A value of *p* < 0.05 was considered statistically significant.

## 5. Conclusions

In summary, we demonstrated that KLF5 is crucial for androgen/AR signaling to activate the transcription of specific genes, including some that mediate cell proliferation, and to promote cell proliferation and tumor growth in PCa cells. Mechanistically, androgen promotes the expression of KLF5 via AR, KLF5 in turn promotes the transcription of AR by binding to AR promoter, and KLF5 and AR coordinate to transactivate the target genes of AR. These findings not only suggest that KLF5 is a crucial factor for the function of AR in PCa cells, they will also further the understanding of how AR signaling is sustained in CRPC. In addition, they will likely facilitate the development of therapeutic strategies for the treatment of CRPC.

## Figures and Tables

**Figure 1 cancers-12-00748-f001:**
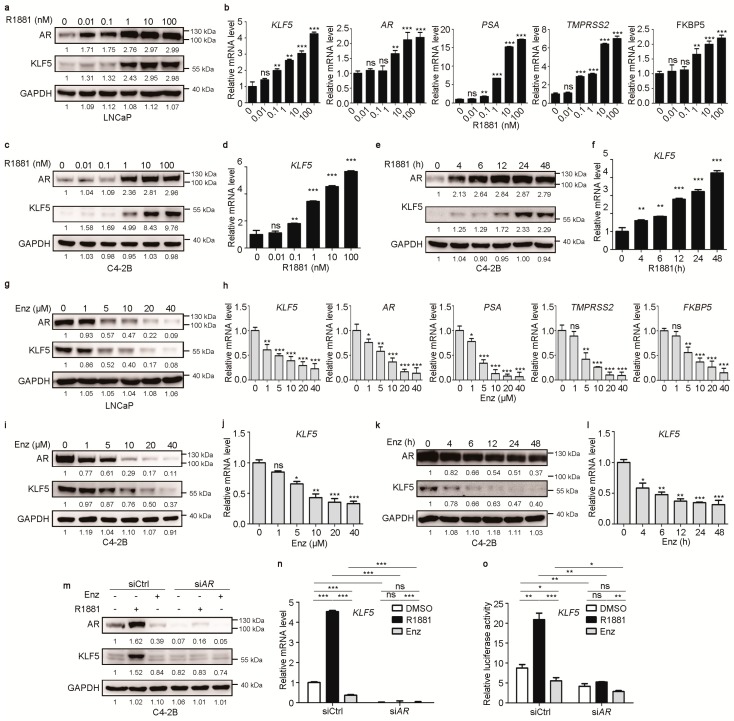
Androgen-androgen receptor (AR) signaling upregulates the transcription of KLF5 in PCa cells. (**a**–**d**) R1881 induced the expression of KLF5 at both protein (a, c) and RNA (b, d) levels in LNCaP (a, b) and C4-2B (c, d) cells. After 24-hour culture in phenol red–free RPMI-1640 medium containing 10% charcoal-stripped (CS) FBS, cells were treated with R1881 for 24 h at the indicated concentrations. Western blotting and real-time qPCR were performed to detect protein and mRNA respectively. (**e**,**f**) R1881 induced the expression of KLF5 at both protein (e) and RNA (f) levels at the indicated times in C4-2B cells. Cell culture conditions and the detection of KLF5 protein and mRNA were the same as in panels a-d. After 24-hour culture, cells were treated with R1881 (10 nM) for the indicated times. (**g**–**j**) Enzalutamide inhibited the expression of KLF5 at both protein (g, i) and RNA (h, j) levels in LNCaP (g, h) and C4-2B (i, j) cells. Cells were cultured in complete media for 24 h and treated with enzalutamide at the indicated concentrations for 24 h. (**k**,**l**) Enzalutamide inhibited the expression of KLF5 at both protein (k) and RNA (l) levels at the indicated times in C4-2B cells. Cell culture conditions and the detection of KLF5 protein and mRNA were the same as in panels g-j. After 24-hour culture, cells were treated with enzalutamide (Enz, 10 µM) for the indicated times. (**m**,**n**) RNAi-mediated silencing of AR prevented R1881 from upregulating KLF5 expression at both protein (m) and mRNA (n) levels in C4-2B cells. Cell culture conditions and the detection of KLF5 protein and mRNA were the same as in panels a-d. Transfection of siRNAs was for 6 h before R1881 treatment (10 nM). Enzalutamide (Enz, 10 µM) was used as a control. siCtrl, control siRNA; siAR, AR siRNA. (**o**) Knockdown of AR also prevented R1881 from inducing transcriptional activity of the KLF5 promoter in C4-2B cells, as detected by the promoter luciferase reporter activity assay. Experimental conditions were the same as in panels i and j except that the reporter plasmid was co-transfected with siRNAs. ns, not significant; *, *p* < 0.05; **, *p* < 0.01; ****p* < 0.001.

**Figure 2 cancers-12-00748-f002:**
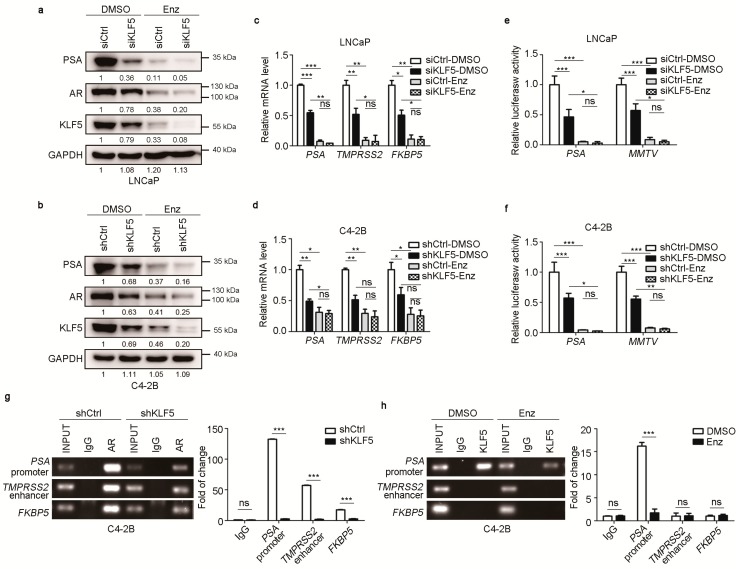
KLF5 is crucial for the transcriptional activity of AR in PCa cells. (**a**–**d**) Knockdown of KLF5 reduced the expression of AR transcriptional target genes *PSA*, *TMPRSS2*, and *FKBP5*. Gene expression was detected for protein by western blotting (a, b) and real-time qPCR for mRNA (c, d). LNCaP (a, c) and C4-2B (b, d) cells in full medium were transfected with siRNAs (a, c) or infected with shRNA lentiviruses (b, d) to silence KLF5. One group of cells were treated with enzalutamide (10 µM, 24 h) to inhibit AR function, which served as a control. siCtrl and shCtrl are control siRNA and shRNA respectively. (**e**,**f**) Knockdown of KLF5 reduced the activities of two androgen-responsive promoters, *PSA* and *MMTV*, in the same cells with the same treatments as in panels a-d, except that the PSA– or MMTV–luciferase reporter plasmid and Renilla-luciferase reporter plasmid were transfected for 24 h before enzalutamide treatment. (**g**) Binding of AR to the promoters of *PSA* and *FKBP5* and the enhancer of *TMPRSS2* was detected after the knockdown of KLF5 in C4-2B cells, as detected by ChIP and regular PCR (left) or real-time qPCR (right). Cells were infected with lentiviruses expressing shRNAs against KLF5 (shKLF5) or control (shCtrl) to knock down KLF5. (**h**) KLF5 binds to the promoter of *PSA* but not the promoter of *FKBP5* or the enhancer of *TMPRSS2* in C4-2B cells in full medium, as detected by ChIP and regular PCR (left) or real-time qPCR (right). Cells were treated with enzalutamide (10 µM, 24 h), with DMSO as a control. ns, not significant; *, *p* < 0.05; **, *p* < 0.01; ****p* < 0.001.

**Figure 3 cancers-12-00748-f003:**
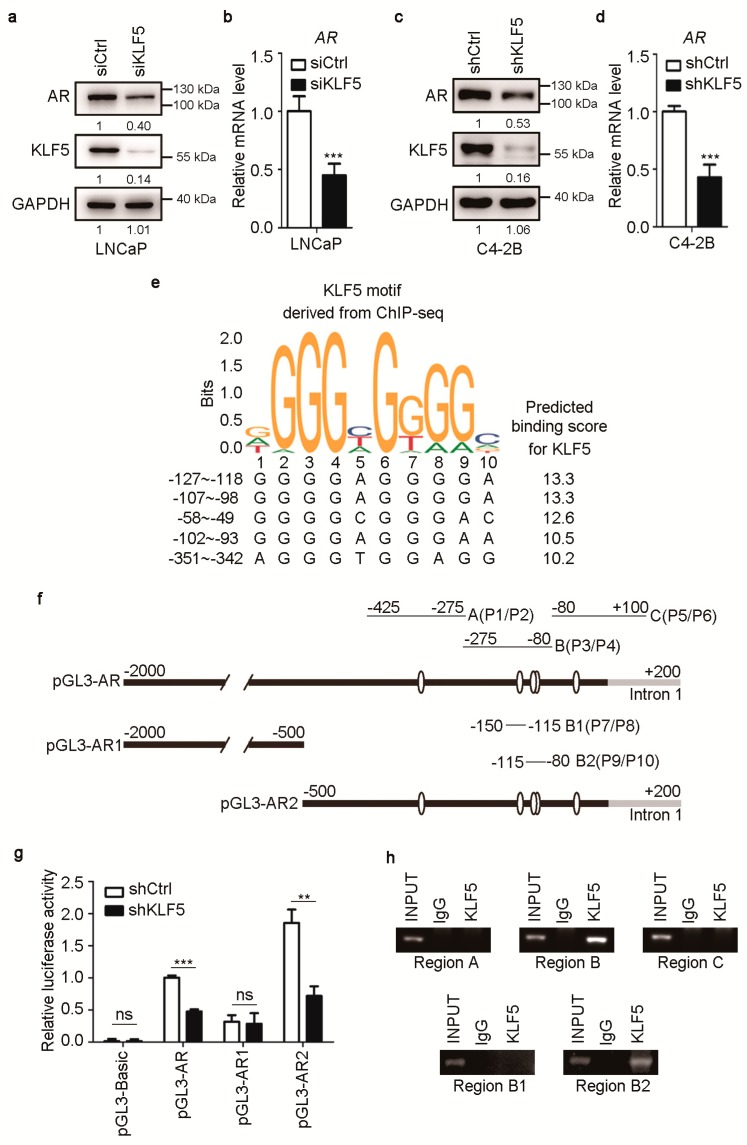
KLF5 is required for the transcription of AR in PCa cells. (**a**–**d**) Knockdown of KLF5 decreased AR expression at both protein (a, c) and RNA (b, d) levels in LNCaP (a-b) and C4-2B cells (c–d). Cells were cultured in complete media for 24 h and transfected with siRNAs for 48 h in LNCaP cells or infected with lentiviruses expressing shRNA targeting KLF5 (shKLF5) in C4-2B cells. Western blotting and real-time qPCR were performed to detect protein and mRNA respectively. (**e**) The AR promoter contains multiple potential KLF5 binding sites, as predicted by aligning the 2-Kb immediate promoter sequence of AR to the consensus KLF5 binding sequence (top) defined in the JASPAR database. Location of these sequences relative to the transcription initiation site (+1) is shown at left. (**f**) Schematic of the AR promoter region (−2000 to +200) with the locations of predicted KLF5 binding sites (empty oval) and primers used for PCR amplification of 5 regions (A, B, C, B1, B2) of the AR promoter spanning the potential binding sites. (**g**) Knockdown of KLF5 decreased AR promoter activity in C4-2B cells, as detected by the luciferase activity assay. The pGL3 vector was used to express full-length AR promoter (g, pGL3-AR, from −2000 to +200) and two shorter AR promoter fragments (pGL3-AR-1, −2000 to −500; pGL3-AR-2, −500 to +200), with their luciferase readings normalized by that of the pGL3-Basic vector control. (**h**) Detection of KLF5-bound AR promoter DNA in C4-2B cells using ChIP and PCR. ns, not significant; *, *p* < 0.05; **, *p* < 0.01; ****p* < 0.001.

**Figure 4 cancers-12-00748-f004:**
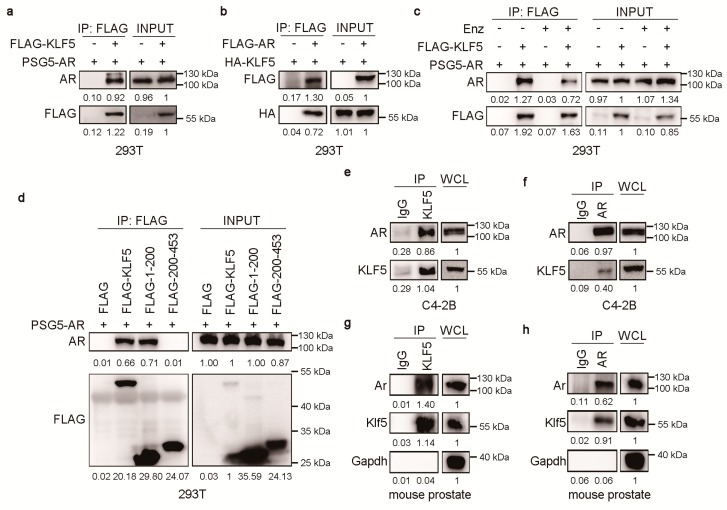
KLF5 physically associates with AR in epithelial cells. (**a**,**b**) HEK293T cells were transiently transfected with expression plasmids of vector control, Flag-tagged (a) or HA-tagged KLF5 (b), and pSG5-AR (a) or FLAG-tagged AR (b), and then subjected to co-IP with FLAG antibody and western blotting with indicated antibodies. (**c**) HEK293T cells transfected with KLF5 and/or AR as in panel a for 24 h were treated with 10 µM enzalutamide for 24 h, and then subjected to co-IP and western blotting with indicated antibodies. (**d**) Mapping of interacting KLF5 regions that interact with AR by co-IP with Flag antibody and western blotting with indicated antibodies in HEK293T cells expressing AR and different fragments of KLF5. (**e**,**f**) Detection of endogenous KLF5-AR association in C4-2B cells by co-IP with KLF5 (e) or AR antibody (f) and western blotting with indicated antibodies. WCL is whole cell lysates before IP. IgG was used as a negative control for co-IP. (**g**,**h**) Detection of the Klf5-Ar association in mouse prostates by co-IP with KLF5 (g) or AR antibody (h) and western blotting with indicated antibodies.

**Figure 5 cancers-12-00748-f005:**
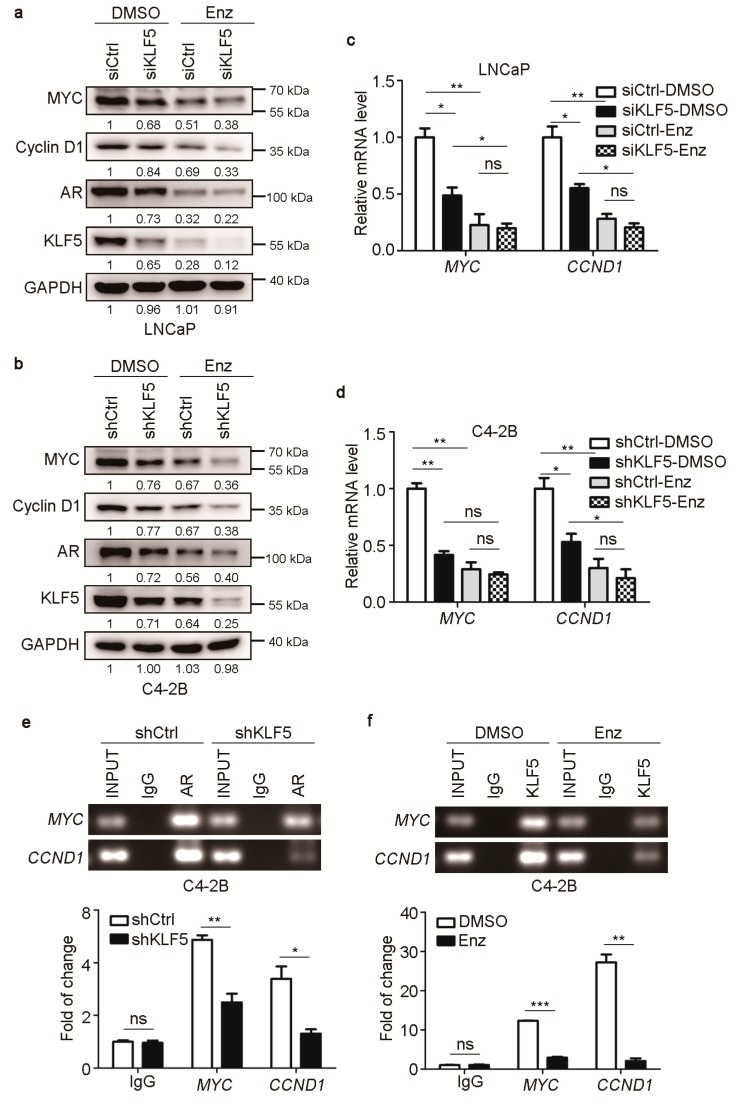
Knockdown of KLF5 attenuates AR-mediated expression of MYC and cyclin D1 in androgen-responsive PCa cells. (**a**–**d**) RNAi-mediated KLF5 silencing attenuated R1881-promoted expression of cyclin D1 and MYC in LNCaP (a) and C4-2B (b) cells, as detected by western blotting for protein (a, b) and by real-time qPCR for mRNA (c, d). One group of cells were treated with 10 μM enzalutamide for 72 h. (**e**) Binding of AR to the promoters of *MYC* and *CCND1* was detected by ChIP and PCR (top) or real-time qPCR (bottom) in C4-2B cells expressing shRNAs against KLF5 (shKLF5) and control (shCtrl). (**f**) Binding of KLF5 to the promoters of *MYC* and *CCND1* was detected by ChIP and PCR (top) or real-time qPCR (bottom) in C4-2B cells treated with or without enzalutamide (10 µM, 24 h). ns, not significant; *, *p* < 0.05; **, *p* < 0.01; ****p* < 0.001.

**Figure 6 cancers-12-00748-f006:**
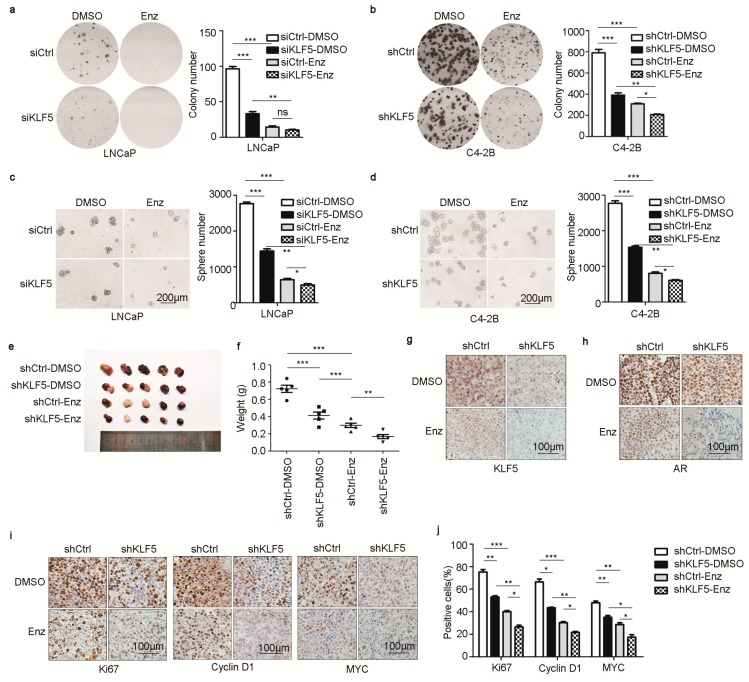
KLF5 is also crucial for androgen/AR signaling to promote cell proliferation and tumor growth in PCa cells. (**a**–**d**) Knockdown of KLF5 by siRNA in LNCaP cells or by shRNA in C4-2B cells reduced colony forming efficiency in 2-D culture (a, b) and sphere formation in Matrigel (c, d). Cells with KLF5 knockdown were seeded onto 6-well plates at 2000 cells/well for LNCaP and 1000 cells/well for C4-2B in regular medium for colony formation assay, and at 4000 cells/well for LNCaP and 2000 cells/well for C4-2B cells for sphere formation assay. Regular media were used, and enzalutamide (10 µM) treatment was applied. The culture time was 2 weeks for C4-2B and 3 weeks for LNCaP in both assays. Images of colonies or spheres were taken (left), and their numbers were counted (right). Only spheres with a diameter greater than 80 µm were counted. Scale bars, 200 μm. (**e**,**f**) Knockdown of KLF5 attenuated tumor growth of C4-2B cells in nude mice, as indicated by tumor images (e) and tumor weights at excision (f). (**g**,**h**) Knockdown of KLF5 reduced AR expression in xenograft tumors of C4-2B cells, as detected by immunohistochemical (IHC) staining with anti-KLF5 (g) and anti-AR (h) antibodies. (**i**–**j**) Knockdown of KLF5 reduced cell proliferation, as indicated by the Ki67 index, and the expression of cyclin D1 and MYC in tumor xenografts of C4-2B cells, as detected by IHC staining (i) and quantitation of positive cells (j). Scale bars, 100 μm. ns, not significant; *, *p* < 0.05; **, *p* < 0.01; ****p* < 0.001.

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
