# Peer review of "KLF5 Is Crucial for Androgen-AR Signaling to Transactivate Genes and Promote Cell Proliferation in Prostate Cancer Cells"

_cancers, 2020, doi:10.3390/cancers12030748_

Round 1
Reviewer 1 Report
Impressive and well done. The authors may discuss how Krüppel-like factor (KLF) family of transcription factors play a role in tumor microenvironment involved in prostate cancer. Another
important unexplored aspect of KLFs is their actions at the crossroad of metabolism and immunity (immunometabolism). The regulatory pathways governing metabolism and inflammation are tightly linked. Visceral obesity induces chronic inflammation within visceral adipose tissue, which in turn contributes to the development of cardiometabolic diseases, such as atherosclerosis and type 2 diabetes, in part by promoting inflammation in the affected tissues. The regulation of cellular metabolism is integral to the inflammatory and regulatory activation of immune cells. It has not yet been well-addressed, but systemic and/or local metabolic disturbances may alter immune cell activities by modulating their cellular metabolism. Metabolism and immunity are thus intricately connected at the cell, tissue and system levels. Future studies will need to address how KLF-dependent regulation of metabolism in various cells types, including immune cells, contribute to inflammatory processes in metabolic tissues and cancer.
Reviewer 2 Report
The manuscript by Li et al., analyze the interconnexion between the transcription factor Krüppel-like factor 5 (KLF5)a and the Androgen receptor and their functional implication in prostate cancer proliferation.
This work is valuable and innovative and could represent an important contribution in the field, but before acceptance some modifications are strongly needed.
Mayors
1) Introduction: Lines 72-74, this sentence is hard to follow. Please, rephrase
2) Results - Fig 1: Why the authors use only Metribolone (methyltrienolone - R1881) as AR agonist? Did you try to perform the experiments of figure 1 with Mibolerone or DHT? It should be interesting to demonstrate how the cells respond to the addition of DHT (for instance).
3) Results – Fig 1: The authors should explain the AR positive feedback that occur when an AR agonist is added or, on the contrary, the negative feedback induced by an antagonist (enzalutamide). At least, a couple of references should be necessary.
4) Result – Fig 1: The authors should check the expression of other AR transcriptional targets (PSA, TMPRSS2, FKBP5….) in these panels as they do in Figure 2.
5) Results – Fig 2 and 5: The bands obtained for KLF5 in figs 2a and 2b and figs 5a and 5b are quite similar. Please try to repeat these WB.
6) Lines 507-510: this sentence is hard to follow. I would rephrase this line
7) The English language need to be corrected in several instances
Minors
Line 18: Please a synonymous for “development”
Line 23: …..C4-2B cells and silencing KLF5, in turn, reduced AR transcriptional…
Line 34: Please define and use the abbreviation “Prostate Cancer (PCa)”
Line 55: Any synonymous for “pathway”?
Lines 54-63: Why do you write “Klf5” and in other lines “KLF5”. Is there a difference ?
All the figures: Please delete “Li et al., Figure x”
Round 2
Reviewer 2 Report
The authors have addressed all my comments/suggestions. I found their responses quite satisfactory and the revised version has been much improved. I now recommend the paper for publication in Cancers